# The Role of the p21-Activated Kinase Family in Tumor Immunity

**DOI:** 10.3390/ijms26083885

**Published:** 2025-04-20

**Authors:** Tianqi Lu, Zijun Huo, Yiran Zhang, Xiaodong Li

**Affiliations:** 1Key Laboratory of Cell Biology, National Health Commission of the PRC and Key Laboratory of Medical Cell Biology, Ministry of Education of the People’s Republic of China, Department of Cell Biology, China Medical University, Shenyang 110122, China; 18304019517@163.com; 2Department of Pharmaceutical Neuroendocrinology, School of Pharmacy, China Medical University, Shenyang 110122, China; 3The Second Clinical College, China Medical University, Shenyang 110122, China; 2021340805@cmu.edu.cn (Z.H.); 2021340712@cmu.edu.cn (Y.Z.)

**Keywords:** PAK, small molecule inhibitor, tumor, immune infiltration

## Abstract

The p21-activated kinases (PAKs) are a group of evolutionarily conserved serine/threonine protein kinases and serve as a downstream target of the small GTPases Rac and Cdc42, both of which belong to the Rho family. PAKs play pivotal roles in various physiological processes, including cytoskeletal rearrangement and cellular signal transduction. Group II PAKs (PAK4-6) are particularly closely linked to human tumors, such as breast and pancreatic cancers, while Group I PAKs (PAK1-3) are indispensable for normal physiological functions such as cardiovascular development and neurogenesis. In recent years, the association of PAKs with diseases like cancer and the rise of small-molecule inhibitors targeting PAKs have attracted significant attention. This article focuses on the analysis of PAKs’ role in tumor progression and immune infiltration, as well as the current small-molecule inhibitors of PAKs and their mechanisms.

## 1. Introduction

### 1.1. PAK Family

The p21-activated kinases (PAKs) belong to the serine/threonine protein kinase family and were initially discovered in mouse brain tissue in 1994. Based on structural variations and sequence similarity, PAKs are categorized into two groups: Group I comprises PAKs 1, 2, and 3, while Group II includes PAKs 4, 5, and 6. Group I PAKs and a subset of Group II PAKs exhibit homology with the STE20 protein family found in yeast, sharing comparable functions in extracellular signaling and reorganization of the cytoskeleton. PAK1 shows significant expression in various adult tissues; PAK2 is predominantly found in endothelial cells; PAK4 exhibits widespread expression across tissues and is notably abundant during embryonic development; and other members of the PAK family are prominently expressed in the nervous system.

### 1.2. PAK Structure

PAKs feature a p21-binding domain (PBD), alternatively referred to as the Cdc42 and Rac1-binding (CRIB) domain, located at their N-terminal region, and a serine/threonine kinase domain at the C-terminal end. Both Group I and Group II PAKs share approximately 50% amino acid identity within these two structural domains. Furthermore, PAKs possess isoform-specific, conserved, proline-rich regions. Notably, the CRIB domain, highly conserved in the regulatory region, interacts with small GTPases from the Rho family, disrupting PAK dimers and inducing activation through reorganization of the kinase’s active site [1,2]. Group I PAKs are distinguished by an autoinhibitory domain (AID) that overlaps with the p21-binding domain (PBD), along with three distinct SH3-binding domains. Among these, two bind to well-known adaptor proteins, Nck and Grb, while the third interacts with β-PIX (also known as ARHGEF7), a PAK interaction exchange factor. There is a significant degree of sequence similarity among Group I PAKs, with more than 88% similarity in the PBD/AID region and over 93% in the kinase structural domain. In vivo, Group I PAKs exist in two states: a self-inhibited PAK dimer where kinase activity is suppressed, and a catalytically active PAK monomer [1,2].

Group II PAKs operate primarily as a monomer, and current understanding proposes two theoretical models to elucidate their activation process. One model suggests the presence of an AID-like domain that initially binds to the kinase structural domain in a self-inhibitory cis configuration. Upon Cdc42 or Rac binding to the GBD (GTPase-binding domain), a structural rearrangement occurs, liberating the kinase domain from its autoinhibitory state.

In contrast, the second model posits a dual-phase activation mechanism. This model involves not only the interaction of Rho GTPases but also the engagement of proteins containing SH3 domains, which interact with AID-like domains. This coordinated interaction further modulates the PAK activation dynamics.

### 1.3. Functions of PAKs

PAKs mainly function by phosphorylating target proteins, thereby regulating the cytoskeleton, cell cycle, and some classic signaling pathways such as the Wnt pathway, PI3K pathway, MAPK pathway, and apoptosis pathway, ultimately promoting cell proliferation or migration (Figure 1).

### 1.4. Role of PAKs in Tumor Development

PAKs play pivotal roles in various critical biological functions, including the modulation of cell growth, survival pathways, cytoskeletal reorganization, and cell migration. However, dysregulated PAK activity has been implicated in cancer pathogenesis. Notably, several types of cancers—such as hepatocellular carcinoma, glioblastoma, breast cancer, esophageal squamous cell carcinoma, colon cancer, bile duct cancer, endometrial cancer, bladder cancer, pancreatic cancer, and non-small-cell lung cancer—are closely associated with PAK regulation. This connection underscores the potential significance of PAKs as a therapeutic target in combating cancer and addressing issues related to drug resistance in cancer treatment (Table 1 and Table 2).

Below, we have compiled a summary of the expression patterns of PAKs in various prevalent cancers and their roles in the progression of these malignancies (Table 3).

## 2. PAKs and Tumor Immunity

### 2.1. Basic Concepts of Tumor Immunity and Tumor Microenvironment

Traditional cancer treatments mainly include surgical resection of tumor tissues and cytotoxic treatments such as chemotherapy, but these methods still have many drawbacks, such as the inability to completely remove the tumor cells, and the lack of targeting can cause damage to normal tissues of the organism. Therefore, in order to be able to effectively kill tumor cells in a targeted manner, increasing attention has been paid to the role of the immune system in tumor therapy.

The development of oncology treatments has been largely due to immunotherapy, which aims to enhance natural defenses to eliminate malignant cells. It represents a huge breakthrough in cancer treatment and has revolutionized the field of oncology. The key targets of tumor immunity are enhancement of immune system reactivity, suppression of immune checkpoints and promotion of tumor cell lysis.

Under normal conditions, the human immune system is capable of recognizing and eliminating tumor cells, thereby reducing their survival rate and playing a protective role in preventing tumor development and progression. However, certain tumors have the ability to alter the host’s immune system, evading its effects, which leads to malignant progression and the failure of cancer immunotherapy. This phenomenon is known as immune evasion by tumors. The mechanisms underlying immune evasion are complex and involve multiple pathways through which tumor cells regulate specific substances to suppress both cellular and humoral immunity, thereby promoting immune escape. Furthermore, changes in the tumor microenvironment (TME) are also of significant importance. Tumor tissues have the capacity to establish a unique microenvironment that supports tumor cell proliferation and survival. Tumors can increase blood supply to the tumor tissue by stimulating angiogenesis, and they can also suppress the functions of various immune cells, for example, by impairing the ability of microglia and macrophages to recognize tumor cells. Additionally, tumors can alter the extracellular matrix (ECM), facilitating invasion. Changes within the tumor microenvironment can interfere with effective anti-tumor immune responses by downregulating or inhibiting the frequency and/or function of immune effector cells and specialized antigen-presenting cells [94]. 

In the tumor microenvironment (TME), immune cell infiltration assumes a pivotal role. These immune cells, including T cells, B cells, neutrophils, and NK cells, form the cornerstone of immunotherapy. Immune cell infiltration represents a crucial aspect of the TME, impacting tumor immunosuppression, distant metastasis, local drug resistance, and the efficacy of targeted therapies. For instance, in colorectal cancer (CRC), higher proportions of infiltrating CD8 and CD4 T cells, particularly Th1 cells, correlate with improved patient prognosis. T-cell-mediated cellular immunity stands as a critical component of human tumor immunity [95,96]. The programmed death receptor 1 (*PD-1*) gene, belonging to the immunoglobulin gene superfamily and associated with classical forms of programmed cell death, has emerged as a focal point in recent research on tumor immunotherapy. PD-1 functions by exerting inhibitory effects on T cell receptor (TCR) signaling. Specifically, when programmed death ligand 1 (PD-L1) binds to PD-1 receptors on T cells, it suppresses T cell proliferation, cytokine production, and cytotoxic activity. This interaction is pivotal in regulating immune responses within the tumor microenvironment [97]. Similarly, it has been reported that high infiltration of tumors by dendritic cells (DC) is associated with a more favorable clinical outcome. DC dendritic cells are known for their plasticity, and when they mature in environments such as tumor microenvironments (TMEs) enriched with immunosuppressive cytokines, they are capable of acquiring tolerogenic phenotypes, which explains the association of DC with a poor prognosis. NK cells are also subject to an immunosuppressive TME influence, participating in the recognition and elimination of tumor cells through cytotoxic activity [98,99]. Tumor-associated macrophages (TAMs) are the most abundant immune population in the TME, and TAMs support cancer cell growth and metastasis, and mediate immunosuppressive effects on TME-adapted immune cells [100]. TAMs are heterogeneous, ranging from anti-tumor to pro-tumor, and anti-tumor TAMs retain antigen-presenting cell (APC) properties, including high MHCII expression, phagocytosis, and tumor-killing activity [101]. Anti-tumor TAMs secrete pro-inflammatory cytokines that support and activate adaptive immune cells, whereas pro-tumorigenic TAMs are immunosuppressive [102,103]. 

### 2.2. PAKs in Tumor Immunity

In tumors, PAKs are associated with uncontrolled cellular proliferation, altered signaling, increased metastasis, drug resistance, and immune system regulation, and can influence tumor development through the tumor microenvironment. As described below, we conducted an analysis of PAK gene expression and immune cell infiltration in various tumor cells using resources from the TCGA database on the TIMER2.0 platform (http://timer.cistrome.org/ (accessed on 1 December 2024) (Figure 2, Figure 3, Figure 4 and Figure 5).

Through a literature review, we ascertained that PAK2 promote the proliferation, migration, and invasion of tumor cells, which was conjectured to be related to the impact of PAK2 on the infiltration of CD4+ T cell immunity [104]. Research indicates that PAK1 functionally regulates the migration of thyroid cancer cells. By inhibiting a group of PAKs using small-molecule inhibitors, a decrease in the vitality, cell cycle progression, migration, and invasion of thyroid cancer cells was achieved. Through the analysis of the aforementioned data, we suggest that this is associated with the immune infiltration of macrophages, particularly M2-type macrophages, by PAKs [105]. Simultaneously, PAK1 is believed to promote the proliferation and invasion of non-small cell lung cancer cells through the ERK pathway. Similarly, we speculate that PAK1 also facilitates proliferation and invasion by reducing the extent of CD8-positive T cell infiltration in non-small cell lung cancer [106].

The role of PAK1 in the tumor microenvironment has garnered considerable attention. Inhibition of PAK1 disrupts immune evasion facilitated by pancreatic stellate cells (PSCs) in pancreatic ductal adenocarcinoma (PDA), thereby concomitantly decreasing inherent PD-L1 expression in PDA cells influenced by both endogenous factors and PSCs [107]. This action sensitizes PDA cells further to CD8+ T-cell cytotoxicity. Inhibiting PAK1 not only diminishes PSC activation and increases tumor-infiltrating lymphocytes (TILs) but also enhances lymphocyte-induced tumor cell death. By downregulating PD-L1, it disrupts the protective role of PSCs, rendering them less resistant to cytotoxic lymphocyte-mediated killing [108]. 

Inhibition of PAK1 diminishes PSC activation, thereby thwarting their protective effects on cancer cells, enabling lymphocytes to eradicate cancer. Furthermore, PAK1 knock out downregulates PD-L1 expression in PDA cells, reinforcing the potency of cytotoxic lymphocyte-mediated killing. PAK1 additionally functions as a specific effector of Rac1, crucial in orchestrating cellular cytoskeletal rearrangements. Positioned upstream of MEK yet downstream of PI3K, PAK1 sequentially orchestrates the signaling pathway pivotal to NK cell cytotoxicity. Activation of PAK1 occurs through β1 integrin crosslinking on NK cells. Identified as a promising target, PAK1 has the potential to hinder excessive proliferation of B lymphocytes in conditions like B cell leukemia [109,110]. 

Additionally, at the genetic level, PAK1 has been found to participate in tumor immunity. Studies have demonstrated a positive correlation between PAK1 expression and immune checkpoint genes (ICPs) in various tumor tissues, suggesting that PAK1 may regulate tumor immune responses through immune checkpoint modulation [111]. PAK2 is involved in regulating the development of regulatory T cells (Tregs), as the loss of PAK2 in T cells results in a significant reduction in thymus- and peripherally-derived Tregs [112]. T cells lacking PAK2 promote the generation of myeloid-derived suppressor cells (MDSCs) by secreting cytokines such as GM-CSF, TNF-α, and IFN-γ. The absence of PAK2 amplifies the immunosuppressive capability of these myeloid suppressor cells by enhancing their proliferation, thereby fostering tumor progression and immune evasion. PAK2 deficiency leads to expanded MDSC populations through intracellular and extracellular pathways, heightening the sensitivity of hematopoietic stem cells to granulocyte-macrophage colony-stimulating factor, diminishing MDSC susceptibility to intrinsic and Fas-Fas ligand-mediated apoptosis, and stimulating CD4 cells to produce increased interferon-γ, tumor necrosis factor, and GM-CSF, thus promoting MDSC expansion [113]. PAK2 amplification is frequently observed in Adult T-cell Leukemia/Lymphoma (ATLL) caused by T-cell lymphotropic virus, promoting CADM1-mediated interactions and enhancing the survival of ATLL cells [114]. Apart from ATLL, significant amplification of PAK is also noted in cutaneous T-cell lymphoma (CTCL), where PAK promotes malignant cell dissemination; PAK kinase inhibitors exhibit specific selectivity towards primary L-CTCL cells with STST3/5 gains, markedly reducing tumor growth and disease dissemination in vivo [115].

Within B lymphocytes, a hierarchical GTPase activation structure is evident, where Rac governs the Ras/MAP kinase pathway’s activity. Research indicates that the Vav/Rac pathway signals independently from Ras to the RB-E2F complex through serine-threonine kinases, PAK2, and PKCε, thereby influencing uncontrolled B cell proliferation.

Furthermore, PAK3 represents a critical signature gene in the original neural subtype of glioblastoma (GBM), impacting its proliferation, differentiation, and growth. Suppression of PAK3 expression in GBM cells alters their differentiation trajectory towards astroglial phenotypes in vitro and promotes tumor progression in vivo [75]. 

Type II PAKs have been found to be significantly expressed at cell-cell junctions, where they primarily regulate adhesion dynamics and colony escape. Inhibiting PAK4 can normalize the tumor vascular microenvironment. Depletion of PAK reduces intratumoral hypoxia and vascular abnormalities, transitioning vascular morphology from characteristics of dilation and tortuosity to normalized vessels. The pivotal role of PAK4 in activating mesenchymal-like transcription within tumor endothelial cells may induce the formation of abnormal vasculature in tumors [116]. Additionally, depletion of PAK4 can increase tumor-specific T cell infiltration, rendering tumors sensitive to PD-1 blockade therapy [117]. 

PAK4 assumes a critical role in mediating immune cell infiltration, where inadequate infiltration is a significant contributor to tumor resistance against therapies. Recent research has highlighted a correlation between heightened PAK4 levels and reduced T cell infiltration. Manipulating PAK4 expression was shown to modulate WNT/β-catenin signaling, augmenting the presence of tumor-infiltrating T cells and improving responsiveness to PD-1 blockade therapy. This finding introduces a novel strategy to enhance the effectiveness of PD-1 blockade therapy [117]. PAK4 additionally stimulates cell proliferation by influencing the activity of immune cells, particularly T cells. Through its interaction with β-catenin, PAK4 phosphorylates and stabilizes β-catenin, augmenting the transcriptional activity of T-cell factor (TCF)/lymphoid enhancer factor (LEF) driven by β-catenin. This cascade ultimately promotes cellular proliferation [118]. In human glioblastomas, PAK4 reduces the expression of intercellular adhesion molecule 1 (ICAM-1) and vascular cell adhesion molecule 1 (VCAM-1) via SLUG, thereby diminishing T cell adhesion to tumor endothelial cells. This inhibition complicates T cell infiltration into tumors and impacts the efficacy of CAR-T immunotherapy in murine models of GBM. Moreover, silencing PAK4 enhances vascular genesis and increases endothelial cell adhesion molecules within the tumor microenvironment, facilitating recruitment of CD8 lymphocytes [119].

Furthermore, from established experiments, it has been observed that PAK6 exerts control over the integrity of adhesion junctions by directly phosphorylating β-catenin in vitro. This phosphorylation event leads to the dissociation of E-cadherin from β-catenin, resulting in diminished cell-to-cell adhesion. Consequently, this process promotes the migration of tumor cells and facilitates evasion from immune surveillance [120]. Moreover, studies have highlighted that PAK6’s ability to target cell-cell adhesion hinges on its N-terminus and necessitates the involvement of its Cdc42/Rac interactive binding (CRIB) domain and the adjacent multi-basic region for achieving optimal targeting efficacy [121]. 

Neutrophils serve as frontline responders to microbial infections and play pivotal roles in various inflammatory conditions within the body. Upon exposure to external stimuli from the tumor microenvironment (TME), neutrophils gather at the lesion site, transitioning between anti-tumor and pro-tumor phenotypes. Anti-tumor neutrophils exert cytotoxic effects directly on tumor cells and indirectly activate adaptive immune responses. Polymorphonuclear neutrophils (PMNs) respond to diverse stress signals by releasing neutrophil extracellular traps (NETs) in a non-apoptotic manner.

PAK is prominently expressed in neutrophils and shows a significant positive correlation with their function. Depletion of PAK has been linked to accelerated tumor growth, suggesting a protective role for PAK in vivo against tumor progression [122].

## 3. PAK and Small-Molecule Inhibitors

The above data reveals the fact that dysregulated PAK expression is usually associated with a variety of human diseases, therefore inhibiting PAKs is a feasible idea with high research value for the treatment of these diseases.

### 3.1. Group I PAK Inhibitors

In the preceding discussion, we examined the pivotal role of the PAK family in the genesis and progression of tumors. The overexpression of PAK1, notably prevalent in various malignant neoplasms, such as pancreatic, breast, and renal cancers, has prompted the development of PAK1-3 inhibitors, which have been demonstrated to ameliorate this condition. However, recent research advancements have unveiled the significant involvement of the PAK family in numerous physiological regulations. For instance, PAK2 regulates the normal development of the cardiovascular and cerebrovascular systems, while PAK3 plays a crucial role in the development of the nervous system and counteracting neurodegenerative diseases. Hindered by these influences, traditional PAK inhibitors such as PF-3758309 [123], FRAX486 [124], and FRAX597 [125], which target the ATP site, exhibit suboptimal selectivity for other PAKs. Pfizer’s evaluation of the PAK inhibitor PF 3758309 in Phase I clinical trials was terminated due to poor drug performance and toxicity [126]. This setback significantly impedes the progress of clinical trials and also underscores the indispensable role of PAK2 in the human body. Notably, upon recognizing this, a plethora of small-molecule inhibitors targeting other sites with better selectivity has emerged (Table 4).

The discovery of IPA3 paved the way for the study of allosteric kinase inhibitors, which differ from pan-PAK inhibitors. IPA3 acts on the regulatory domain of PAK1 to prevent binding of Cdc42, hindering downstream pathways, thereby achieving superior selectivity [127]. While not extensively utilized in tumor research, IPA-3, serving as a direct non-ATP competitive PAK1 inhibitor, has garnered acclaim in studies pertaining to cellular immune-inflammatory diseases and PAK1-mediated cardiac cell development [128,129]. However, its efficacy limitations have hampered further advancements. Subsequently, NVS-PAK1-1 emerged, demonstrating superior inhibition of PAK1 over other PAK isoforms and kinase groups, with high selectivity. Subsequent studies have confirmed its efficacy in combination therapy, effectively improving resistance and prognosis [130]. 

Furthermore, NVS-PAK1-1 diminishes tumor formation, diminishes the average size of dorsal root ganglia (DRG), and extends lifespan in NF2-cko mice. Additionally, it has been observed that, through combination therapy, resistance of ER+ breast cancer to endocrine therapy and CDK 4/6 inhibitors can be overcome. Regrettably, its efficacy remains suboptimal [8,131]. 

On this basis, BJG-05-039 emerged as a PAK1-selective degrader composed of NVS-PAK1-1 conjugated to lenalidomide. It achieves better pharmacological effects by targeting the degradation process of PAK1 and has opened new avenues for inhibitor research [132]. Regarding FRAX1036, its inhibitory effect on breast cancer cells has been validated. It has been selected and widely used for its high potency and moderate kinase selectivity, and it has been demonstrated that its application in breast cancer induces apoptosis and enhances the efficacy of microtubule stabilizing agents, presenting substantial opportunities in the realm of combination therapy. FRAX1036 has faced challenges due to the basic amine part of its compound structure lacking specific interactions [133]. To address this, the selective PAK1 inhibitor G-5555 was designed and exhibits superior oral exposure and bioavailability in animal experiments compared to FRAX1036. Research has revealed its commendable ameliorative effects in the treatment of thyroid tumors and myeloproliferative neoplasms [105,134]. 

Additionally, CP 734 inhibits tumor growth by acting on the V342 residue to inhibit ATP activity and downstream signaling pathways. Its combined application with gemcitabine or 5-fluorouracil has been found to synergistically inhibit the proliferation of pancreatic cancer cells [135].

In addition to designing small molecule inhibitors targeting PAK, natural extracts have also been one of the research targets. Components found in propolis have demonstrated the ability to inhibit PAK1-dependent growth in A549 lung and pancreatic cancer cells. One such component is CAPE (caffeic acid phenethyl ester), which reduces the level of RAC1 GTPase protein and its activity. Consequently, PAK1 is inactivated through the downregulation of NADPH oxidase activity [136]. Conversely, a component found in Brazilian green propolis, known as artepillin C (ARC), may block the PAK1 signaling selectively, without affecting another kinase known as AKT, resulting in the autophosphorylation of PAK1 on Ser141 [137]. 

In the realm of PAK group functionalities, we have elucidated the established correlation of PAK4 with various malignant neoplasms, as well as its roles in shielding cells from apoptosis, fostering cellular motility, and impeding adhesion and anchorage-independent proliferation. All These promote the growth and invasion of tumors, thereby directly diminishing patient prognosis. To impede the effects of PAK4 and its downstream cascades, endeavors spanning several decades have witnessed the trial and development of numerous inhibitors. However, the exigencies of specificity and PAK4’s intrinsic involvement in mammalian cardiovascular development, immune defense establishment, and neuroprotection have hindered the pace of developmental advancement.

**Table 4 ijms-26-03885-t004:** Group I PAK inhibitors.

Inhibitors	Discovery Time	Action Site	Biological Activity
IPA3	2008	PAK1 regulatory domain	IPA-3 is a selective non-ATP competitive PAK1 inhibitor with IC50 of 2.5 μM, and shows no inhibition to group II PAKs
FRAX597	2013	M344 gatekeeper residue and the C-helix in PAK1	FRAX597 is determined to be a potent, ATP-competitive inhibitor of Group I PAKs (PAK 1-3), with IC_50_ of 8, 13 and 19 nM for PAK1, 2 and 3.
FRAX486	2013	ATP pocket of PAK1	Inhibition of PAK1/2 phosphorylation, with IC_50_s of 14, 33 and 39 nM for PAK1, 2 and 3.
NVS-PAK1-1	2015	S144 of PAK1	NVS-PAK1-1 is a potent and selective allosteric PAK1 inhibitor, with an IC_50_ of 5 nM.
FRAX1036	2015	S338 of PAK1	FRAX1036 is a PAK inhibitor, with K_i_s of 23.3 nM, 72.4 nM, and 2.4 μM for PAK1, PAK2 and PAK4.
G-5555	2015	K299/M344/D398 of PAK1	G-5555 inhibits the phosphorylation of MEK1(S298), a downstream substrate of PAK1/2, with K_i_s of 3.7 nM and 11 nM for PAK1 and PAK2.
ARC (Artemisinin C)	2015	Rac/Cdc42	ARC may block PAK1 activation induced by Rac/Cdc42.
G9791	2016	K299/E315 of PAK1	A poyridone side chain analogue with K_i_s values of 0.95 nM and 2.0 nM for PAK1 and PAK2, respectively
CAPE (Caffeic Acid Phenethyl Ester)	2017	Rac1	Inactivation of PAK1 by down-regulating the activity of NADPH oxidase.
CP734	2020	V342 of PAK1	CP734 targets the residue V342 of PAK1 and inhibits its ATP activity with an IC_50_ value of 15.27 μM and without significant inhibitory effect on PAK2, PAK3 or PAK6 [135].
BJG-05-039	2022	S298 of PAK1	BJG-05-039 inhibits the phosphorylation of MEK S298, thereby reducing the activity of PAK1 [132].

### 3.2. Group II PAK Inhibitors

Presently, only PF-3758309 and KPT9274 have entered phase I clinical trials. They are diminutive molecular inhibitors targeting PAK4, attenuating its activity by diminishing PAK4 phosphorylation and subsequent activation of downstream β-catenin/β-PIX. Notably, KPT9274 markedly elevates the population of cells in the G1 phase, reduces the proportion of cells in the S phase, and induces cell cycle arrest in the G1-S transition. Conversely, PF-3758309 exerts minimal influence on the distribution of G1-phase cells but diminishes the population of S-phase cells while augmenting the G2-phase population, indicating cell cycle arrest in the G2/M phase. Both inhibitors dampen PAK4 activity, thereby mitigating molecular signaling implicated in the cell cycle and metastatic progression, downregulating pivotal signaling pathways involved in tumorigenesis and progression, and consequently fostering potent anti-tumor and metastatic attributes [138,139]. However, the former, impeded by its ATP site selectivity, fails to strictly qualify as a PAK4 inhibitor but rather as a pan-PAK inhibitor. Consequently, it exerts equally potent inhibitory effects on type 1 PAK, leading to adverse outcomes and prompting Pfizer to discontinue its development (Table 5).

As for the latter, KPT9274, it stands as a novel dual PAK4/NAMPT modulator, belonging to the domain of allosteric inhibitors. Its initiation into phase I clinical trials in conjunction with anti-PD-1 antibodies has exhibited promise in augmenting anti-tumor efficacy and combating resistance to PD-1 antibodies in patients. Its efficacy has been validated in advanced solid tumors or non-Hodgkin lymphoma. Regrettably, its binding targets and mechanisms of action remain elusive, posing challenges to its clinical application in combination therapy [117,139].

GL-1196 and LC-0882 are both small-molecule inhibitors of PAK4, both of which inhibit the phosphorylation of PAK4, and inhibit the G1/S transition of the gastric cancer cells by downregulating Cyclin D1. In addition, as the kinase activity of PAK4 is inhibited by two inhibitors, the downstream LIMK1/cofilin signaling pathway of PAK4 is also suppressed, ultimately inhibiting the migration and invasion of gastric cancer cells [140,141]. 

LCH-7749944, possessing moderate potency, acts as an inhibitor targeting PAK4, exerting its inhibitory prowess on the migration and proliferation of human gastric cancer cells. It accomplishes this feat by impeding both the PAK4/LIMK1/cofilin and PAK4/MEK-1/ERK1/2/MMP2 pathways. This comprehensive blockade effectively stifles PAK4 activity while also perturbing the plasticity of gastric cell lines, thereby inhibiting filopodia formation. Based on its dual effect on the conserved residues Leu 398 (PAK4) and Leu 347 (PAK1) within the hinge region, subsequent studies have showcased the inhibitory potential of LCH-7749944 against the invasive metastasis of gastric cancer cells [142]. To enhance its specificity towards PAK4, efforts have been made to modify its molecular structure, resulting in CZH226, one of the most selective inhibitors of PAK4 available today. This compound exhibits favorable kinase selectivity, both in vitro and in vivo, alongside advantageous physicochemical properties. By impeding downstream molecular signaling mediated by PAK4, CZH226 effectively suppresses tumor cell migration and invasion. It holds promise as a pioneering agent in the ongoing development of targeted therapeutics against PAK4 [143].

Through structure-based virtual screening, researchers have unearthed a novel, selective small-molecule scaffold, SPU-106, as an inhibitor targeting PAK4. This compound adeptly binds to the C-terminal kinase structural domain of PAK4, thwarting its phosphorylation activity. Its efficacy lies in the effective suppression of SGC7901 gastric cancer cell invasion by attenuating the phosphorylation levels of PAK4 and its downstream effector, SCG10 [144].

The development of PAK inhibitors has been underway for decades. While the causal relationship between PAK1/PAK4 and most malignant tumors has been unequivocally established, the indispensable role of the PAK family in maintaining normal physiological activities, coupled with the high similarity among family member sites, means that potent inhibition while maintaining high selectivity towards PAK1 and PAK4 remains a formidable challenge faced by every researcher. From the classical pan-PAK inhibitor PF-3758309 to the current highly PAK4-selective compound Czh226, countless researchers have contributed to these developments. We expect that more effective PAK4 inhibitors will be developed and applied in clinical cancer treatment in the future.

## 4. Conclusions and Outlook

Since its discovery in 1994, the PAK family has emerged as a prominent molecular entity in cancer research, and its physiological and oncogenic roles have been continuously explored by previous scholars. In this article, we outline and demonstrate the significant achievements and immense developmental potential of PAK in various domains, reviewing its fundamental functionalities, oncogenic properties, and implications in tumor immunology. Based on the preceding discussion, it is evident that proteins within the PAK family not only orchestrate a plethora of pivotal functions within organisms, regulating cellular operations to sustain normal metabolic processes, but also serve as significant factors implicated in the pathogenesis of diverse ailments. In recent years, a burgeoning body of evidence underscores the pivotal clinical significance of PAKs in modulating myriad normal physiological activities across various systems, including cardiovascular, neural, immune, and embryonic domains, encompassing processes such as proliferation, migration, survival, and more (Figure 6).

For instance, the inhibition of PAK1 precipitates neural impairments, including loss of cognitive function and memory, and also results in cardiovascular disorders. In addition, targeting PAK1 in infections attenuates viral damage, albeit at the potential cost of specific immune functions. But such interventions also exert positive effects in acute allergic responses, in which PAK1 regulates mast cell degranulation via effects on calcium mobilization and cytoskeletal dynamics. Similarly, PAK4 orchestrates pivotal cellular processes, regulating proliferation, survival, invasion, metastasis, epithelial–mesenchymal transition, and drug resistance, thereby propelling the overarching progression of cancer. Yet, it also shoulders responsibilities in safeguarding neuronal integrity in Parkinson’s disease and maintaining immune homeostasis. However, the journey of PAK4 inhibitor exploration in both preclinical and clinical arenas has been marred by setbacks due to issues like poor selectivity, cellular functionality, or pharmacokinetic challenges, leading to disappointments. Balancing the utilization of PAK inhibitors for therapeutic purposes while preserving the essential roles of the PAK family in normal physiological function poses a significant challenge in the current landscape of small-molecule inhibitor research and development.

At the same time, there has been an increasing focus on clarifying the roles of PAKs in various human malignancies, with a particular emphasis on their implications in tumorigenesis and the underlying mechanisms at work. Efforts have intensified in the search for small-molecule inhibitors capable of selectively targeting PAKs for the treatment and prevention of cancer. Through a meticulous review of the literature, it has come to light that the PAK family also exerts influence on the extent of immune infiltration within tumor tissues, a factor intimately intertwined with the prognosis of individuals afflicted by such malignancies. Notably, within the realm of Class I and Class II PAK inhibitors, current discourse predominantly revolves around targeted interventions aimed at PAK1 and PAK4. This focus is substantiated by their heightened involvement across a spectrum of human cancers.

The diverse modes of action displayed by each category of PAK inhibitors underscore the significance of delving into strategies to manipulate PAK activity through synergistic drug regimens to attain maximal anti-cancer efficacy while maintaining normal physiological functions. However, the journey towards obtaining market approval remains arduous and protracted. By summarizing the functions of PAK4 in tumors and its existing small-molecule inhibitors, it may help us gain insights for the development of highly specific inhibitors that target its cancer-promoting functions. These initiatives represent promising pathways for future exploration and advancement.

## Figures and Tables

**Figure 1 ijms-26-03885-f001:**
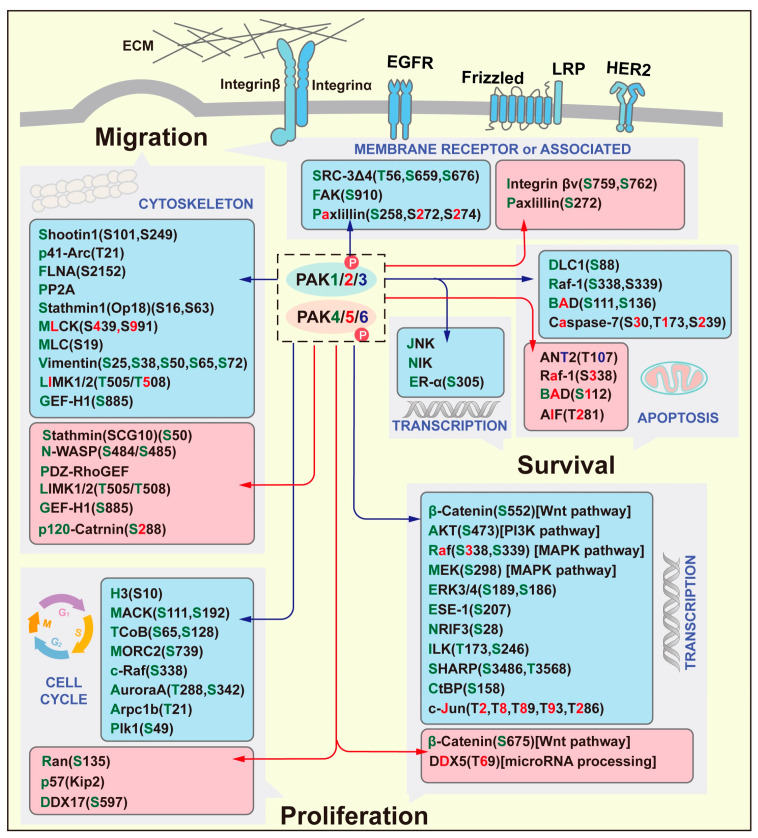
The Functions of PAKs. PAK target proteins that are involved in the process of cell proliferation, migration and survival are listed in the boxes. The targets labeled with green letters represent the phosphorylation by PAK1/PAK4, red letters represent that by PAK2/PAK5, and blue letters represent that by PAK3/PAK6. The numbers of “S/T” in the brackets represent the phosphorylation sites.

**Figure 2 ijms-26-03885-f002:**
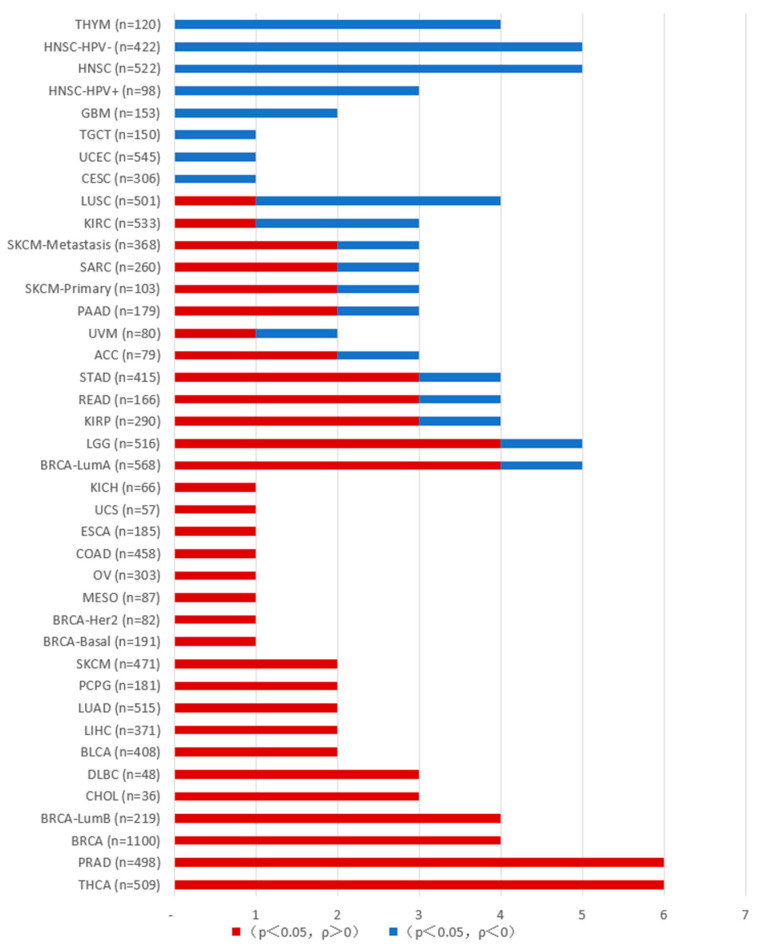
The correlation between PAK and T cell CD8+ infiltration in various tumors. The data were obtained from the Timer platform. When *p* > 0.05, there is no correlation between the gene expression of the PAK family and the degree of T cell CD8+ infiltration in cancer; values marked in red (*p* < 0.05, ρ > 0) indicate a positive correlation between PAK and T cell CD8+ infiltration in cancer; values marked in blue (*p* < 0.05, ρ < 0) indicate a negative correlation between PAK and T cell CD8+ infiltration in cancer.

**Figure 3 ijms-26-03885-f003:**
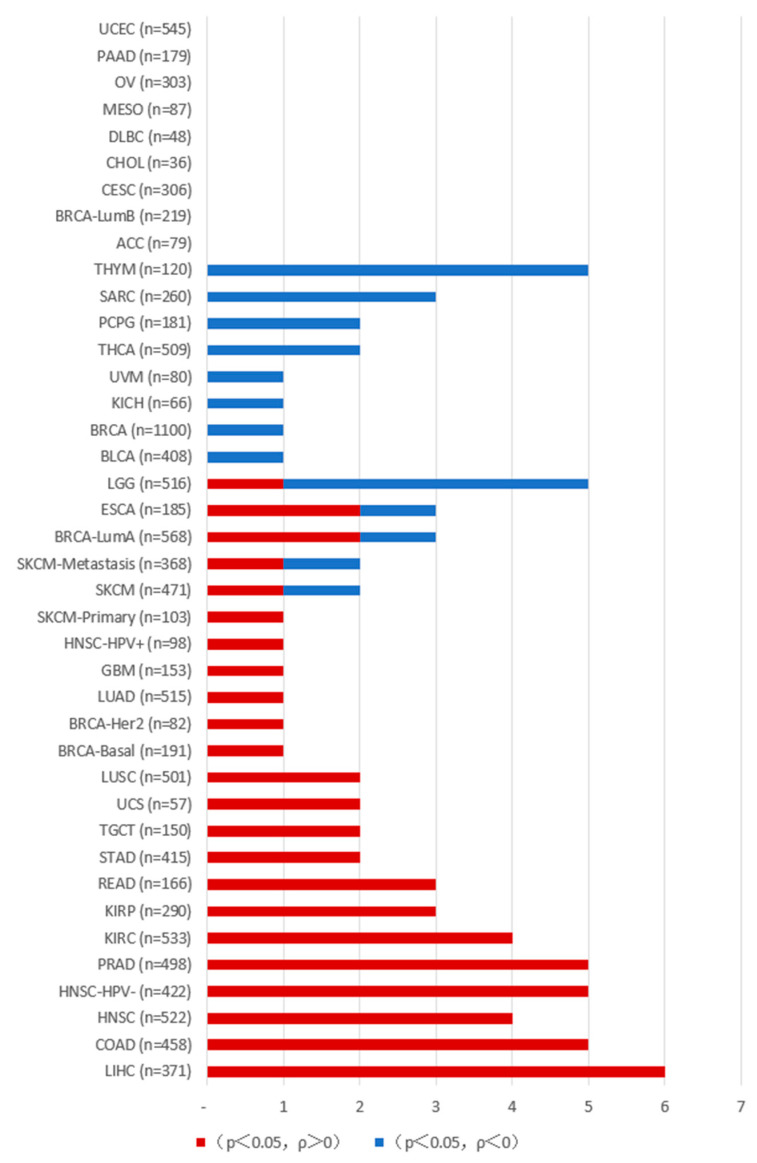
The correlation between PAK and T cell CD4+ infiltration in various types of tumors. The data were obtained from the Timer platform. When *p* > 0.05, there is no correlation between the gene expression of the PAK family and the degree of T cell CD4+ infiltration in cancer; values marked in red (*p* < 0.05, ρ > 0) indicate a positive correlation between PAK and T cell CD4+ infiltration in cancer; values marked in blue (*p* < 0.05, ρ < 0) indicate a negative correlation between PAK and T cell CD4+ infiltration in cancer.

**Figure 4 ijms-26-03885-f004:**
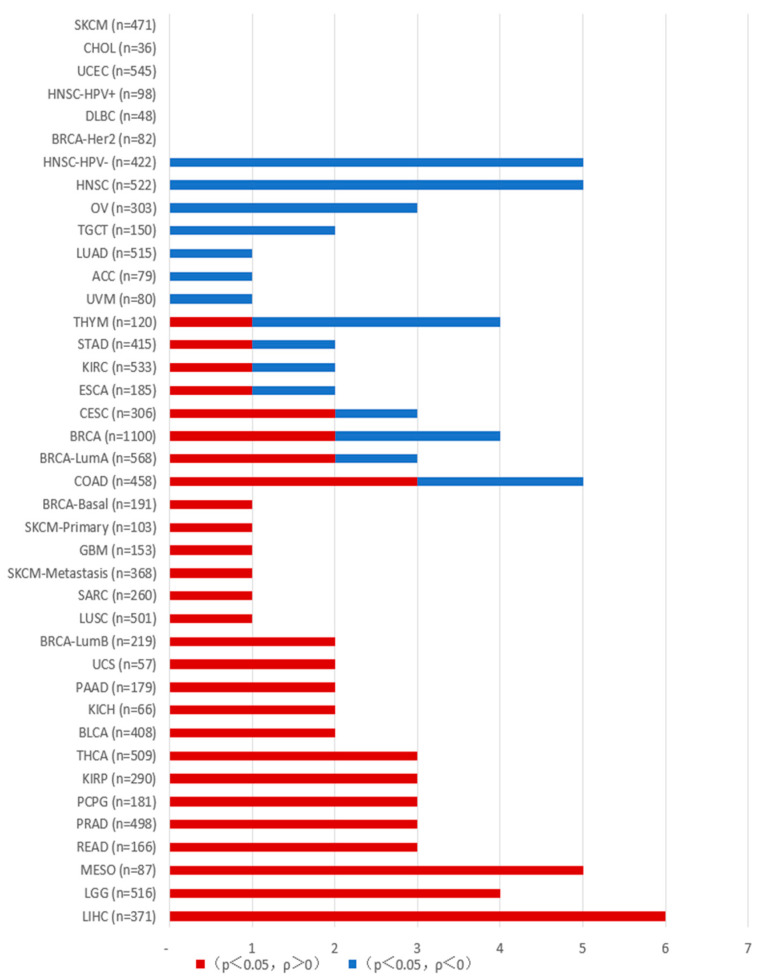
The correlation between PAK and B cell infiltration in various types of tumors. The data were obtained from the Timer platform. When *p* > 0.05, there is no correlation between the gene expression of the PAK family and the degree of B cell infiltration in cancer; values marked in red (*p* < 0.05, ρ > 0) indicate a positive correlation between PAK and B cell infiltration in cancer; values marked in blue (*p* < 0.05, ρ < 0) indicate a negative correlation between PAK and B cell infiltration in cancer.

**Figure 5 ijms-26-03885-f005:**
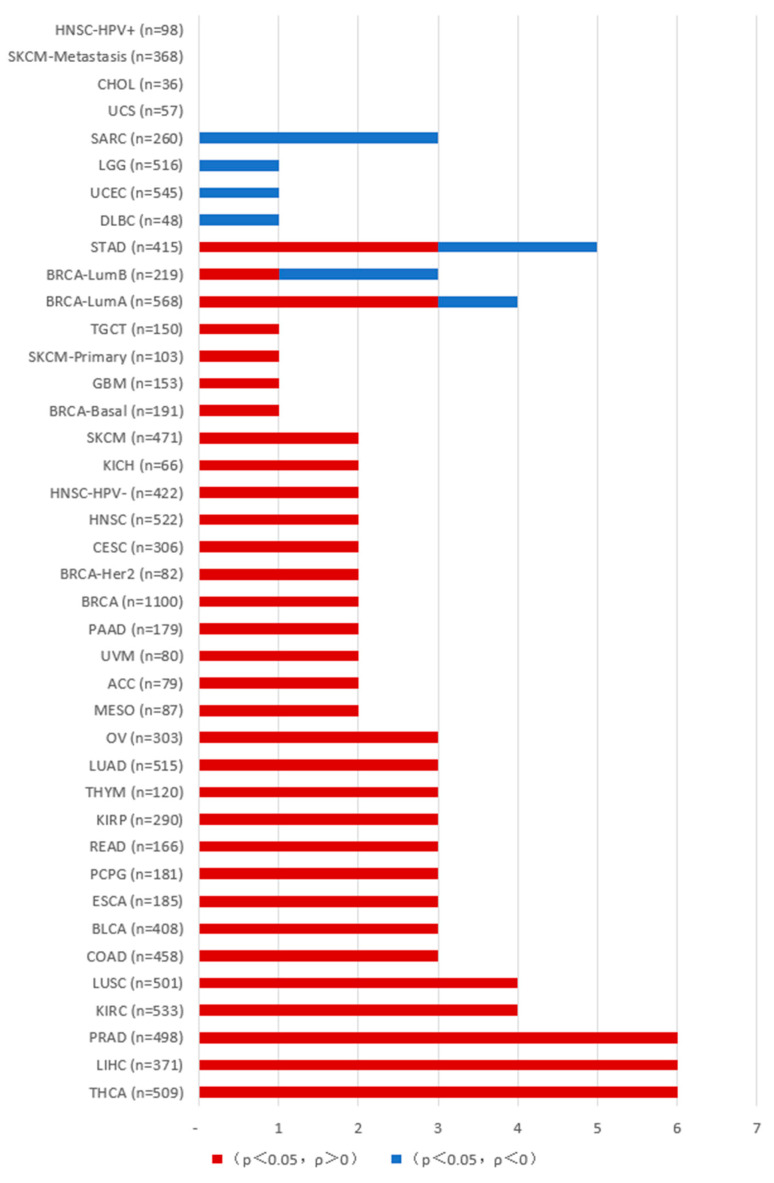
The correlation between PAK and macrophage infiltration in various tumors. The data were obtained from the Timer platform. When *p* > 0.05, there is no correlation between the gene expression of the PAK family and the degree of macrophage infiltration in cancer; values marked in red (*p* < 0.05, ρ > 0) indicate a positive correlation between PAK and macrophage infiltration in cancer; values marked in blue (*p* < 0.05, ρ < 0) indicate a negative correlation between PAK and macrophage infiltration in cancer.

**Figure 6 ijms-26-03885-f006:**
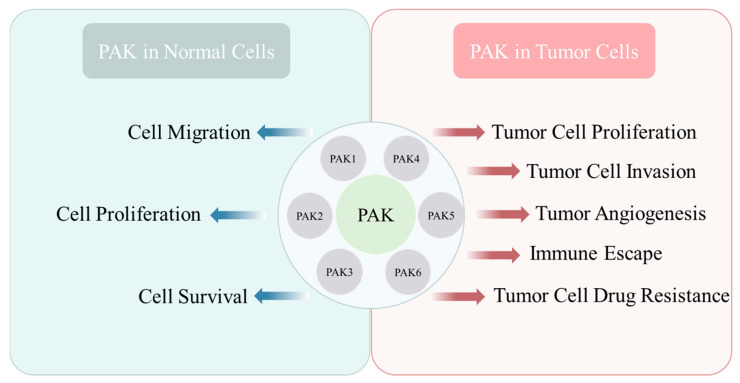
The roles of PAKs in both the normal physiological environment and the tumor environment.

**Table 1 ijms-26-03885-t001:** Role of GROUP I PAKs in tumor development.

Group I PAKs	Functionality	Molecular Pathways
PAK1	Proliferation	PAK1/PI3K/Akt [3] PAK1/LIMK1 [4] PAK1/RAF/MEK/ERK [5,6] PAK1/Wnt/β-catenin/CyclinD1 [7]
	Invasion	PAK1/MMP [7] PAK1/MEK1 [8] PAK1/RUFY3 [9]
	Survival	PAK1/NF-κB [10] PAK1/Bim [11] PAK1/HIF-1α [12] PAK1/FOXO3 [13]
	Angiogenesis	PAK1/β-catenin [14] PAK1/p38/MMP-2 [15] PAK1/STAT5B/CSF2 [16]
PAK2	Proliferation	PAK2/ERK [17] PAK2/MAPK [18]
	Invasion	PAK2/RhoA [19] PAK2/SOX2/DEK [20] PAK2/Cdc42, Rac1 [21] PAK2/LIMK1/cofilin [22] PAK2/PYK2 [23]
	Survival	PAK2/AMPK/YAP [24] PAK2/MCM7 [25]
	Angiogenesis	PAK2/TGF-β [26]
PAK3	Invasion	EMT [27]

**Table 2 ijms-26-03885-t002:** Role of GROUP II PAKs in tumor development.

Group II PAKs	Functionality	Molecular Pathways
PAK4	Proliferation	PAK4/β-catenin/c-Myc [28] PAK4/MEK/ERK [28] PAK4/PI3K/AKT/mTOR [29] PAK4/c-Src/EGFR/CyclinD1 [30] PAK4/LIMK [31] PAK4/ERα [32] PAK4/RELB [33] PAK4/PI3K/Akt [34] PAK4/FH [35] PAK4/Mdm2/p53/G6PD [36] PAK4/Smad2, Smad3/TGF-β [37]
	Invasion	PAK4/PI3K/AKT/mTOR [30] PAK4/LIMK/Cofilin [32] PAK4/PPARγ/Nox1 [38] PAK4/P53 [36,39]
	Survival	PAK4/caspase 8 [40] PAK4/Bad [41]
	Angiogenesis	PAK4/ERK/MMP-2 [42]
PAK5	Proliferation	PAK5/DNPEP/USP4 [43] PAK5/AIF [44] PAK5/DDX5 [45] PAK5/Cyclin D1, β-catenin [46] PAK5/CDK2, CDC25A, Cyclin D1 [47] PAK5/ERK [48]
	Invasion	PAK5/DNPEP/USP4 [43] PAK5/PI3K/AKT [49] PAK5/STATB1 [50] PAK5/DDX5 [45] PAK5/Egr1/MMP2 [51] PAK5/E47 [52]
	Survival	PAK5/AIF [44] PAK5/Bad [53]
PAK6	Proliferation	PAK6/Eg5 [54] PAK6/Mdm2 [55] PAK6/WNT/β-catenin [56]
	Invasion	PAK6/IQGAP3/RhoA [57] PAK6/IQGAP1/E-cadherin/β-catenin [58] PAK6/cofilin [59]
	Survival	PAK6/SIRT4/ANT2 [60] PAK6/ATR/CHK1 [61]

**Table 3 ijms-26-03885-t003:** The expression of PAKs in tumor development.

**Group I PAKs**	**Gene Expression Status**	**Access to Cancers**
PAK1	+	Bladder cancer (↑) [62]
	+	Breast cancer (↑) [63,64]
	+	Gastric cancer (↑) [65,66]
	+	Liver cancer (↑) [67]
	+	Ovarian cancer (↑) [16,68]
	+	Renal cell carcinoma (↑) [66,69]
PAK2	+	Ovarian cancer (↑) [70]
	+	Gastric cancer (↑) [18,71]
	+	Pancreatic cancer (↑) [72,73]
	+	Colorectal cancer (↑) [19]
PAK3	+	Hepatocellular carcinoma (↑) [27]
	−	Cervical cancer (↓) [74]
	−	Glioma(↓) [75,76]
	+	Pancreatic cancer (↑) [72]
**Group II PAKs**	**Gene Expression Status**	**Access to Cancers**
PAK4	+	Breast cancer (↑) [33,77,78]
	+	Pancreatic cancer (↑) [79,80]
	+	Ovarian cancer (↑) [30]
	+	Gallbladder cancer (↑) [81]
	+	Gastric cancer (↑) [37,82]
	+	Hepatocellular cancer (↑) [39,83]
	+	Sarcomas (↑) [84,85]
	+	Endometrial cancer (↑) *
	+	Melanoma (↑) [86,87]
PAK5	+	Breast cancer (↑) [43,44]
	+	Colorectal cancer (↑) [52,88]
	+	Cervical cancer (↑) [50,89]
	+	Lung cancer (↑) [90,91]
	+	Ovarian cancer (↑) *
	+	Osteosarcoma (↑) [92]
PAK6	+	Prostate cancer (↓) [60,93]
	+	Colorectal cancer (↑) [59]
	+	Renal cancer (↓) [66]
	−	Hepatocellular cancer (↑) [54,56]

The content marked with * originates from the TCGA database. “+” in the gene expression status indicates an increase in PAK expression, while “−” indicates a decrease in PAK expression. “↑” represents that PAK promotes the occurrence and development of cancer, while “↓” represents that PAK inhibits the occurrence and development of cancer.

**Table 5 ijms-26-03885-t005:** Group II PAK inhibitors.

Inhibitors	Discovery Time	Action Site	Biological Activity
PF-3758309	2010	C502/L472 of PAK4	Inhibition of all PAKs with an IC_50_ of 39nM to PAK1 and 15nM to PAK4.
LCH-7749944	2012	PAK4 ATP binding pocket	Acts as an ATP-competitive inhibitor. LCH-7749944 is a potent PAK4 inhibitor with an IC_50_ of 14.93 μM.
KPT9274	2014	PAK4 kinase domain	An orally bioavailable, dual PAK4/Nicotinamide phosphoribosyl transferase (Nampt) inhibitor, with IC_50_s of <100 nM and 120 nM.
GNE-2861	2015	DFG-out pocket	GNE-2861 inhibits PAK4, PAK5 and PAK6 with IC_50_s of 7.5, 36, 126 nM.
GL-1196	2016	L398\A348\K350\L447\V335\A402\G330\S331 of PAK4	Inhibition of PAK4.
LC-0882	2017	L398/A348/K350/A492/I327/V335 of PAK4	Inhibition of PAK4.
CZH226	2017	E396/L398/M395/V335/D458 of PAK4	A potent and selective PAK4 inhibitor (PAK4 Ki = 9 nM; PAK1 Ki = 3112 nM).

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
