# Peer review of "The Role of the p21-Activated Kinase Family in Tumor Immunity"

_ijms, 2025, doi:10.3390/ijms26083885_

Round 1
Reviewer 1 Report
Comments and Suggestions for Authors
The review entitled "The role of PAK family in tumor immunity" by Tianqi Lu et al. provides a comprehensive analysis of the role of the PAK family in tumor immunity. The manuscript is very detailed and well documented; however, there are some aspects that could be improved to enhance clarity and readability.
- Formatting of tables and figures. Improve the table readability. Some tables appear too dense and are difficult to interpret. It is recommended to do so:
- Optimize formatting for a better spatial arrangement;
- Consider splitting particularly complex tables into smaller, more manageable tables;
- Ensure that headings are clearly visible and consistent throughout the manuscript.
- More descriptive captions for the figures:
- Some captions are too short and do not adequately explain the content of the figures. We suggest making them more detailed by including a summary of the meaning of the data in the data;
- For graphs showing correlations (e.g. between PAK expression and immune infiltration), it would be useful better to clarify the meaning of the colours and coefficients indicated.
3. Numbering and references to figures:
- Check that the numbering of the figures matches the text. Sometimes, the figures are mentioned with different numbers or without a clear description;
- Ensure that there is an explicit reference in the text to each figure and table used.
4. Clarify whether the analysis is your own or based on existing studies.
- If the authors conducted the analysis, they must explain the methodology, indicating the database used, the criteria for data selection and the statistical approach. Indicate in the legend who the data originated from. The data must be clearly cited if they originate from other publications.
Author Response
1. Summary |
|
|
We are deeply grateful for your insightful comments on our manuscript and extend our most respect for your expertise. As you rightly pointed out, there are several issues within our article that require addressing. In accordance with your suggestions, we have made revisions to the previous draft. The reviewers' comments are listed below in blue, with specific issues numbered. Our responses are provided in regular black font.
|
||
2. Questions for General Evaluation |
Reviewer’s Evaluation |
Response and Revisions |
Is the work a significant contribution to the field? |
|
|
Is the work well organized and comprehensively described? |
|
|
Is the work scientifically sound and not misleading? |
|
|
Are there appropriate and adequate references to related and previous work? |
|
|
Is the English used correct and readable? |
|
|
3. Point-by-point response to Comments and Suggestions for Authors |
|
|
1. Comments 1: Formatting of tables and figures. readability. Some tables appear too dense and are difficult to interpret. lt is recommended to do so:Optimize formatting for a better spatial arrangement; Consider splitting particularly complex tables into smaller, more manageable tables; Ensure that headings are clearly visible and consistent throughout the manuscript. |
||
Response 1: We are truly appreciative of the suggestion put forward by the reviewers. To enhance the clarity and comprehensibility of the manuscript, we have reorganized the original Tables 1 and 2 into Figure 1. Additionally, we have refined the remaining tables by adjusting column widths, line spacing, and text alignment for improved presentation. This change can be found-page3 section3 figure1.
|
||
2. Comments 2: More descriptive captions for the figures. Some captions are too short and do not adequately explain the content of the figures. We suggest making them more detailed by including a summary of the meaning of the data in the data; For graphs showing correlations (e.g. between PAK expression and immune infiltration), it would be useful better to clarify the meaning of the colours and coefficients indicated. |
||
Response 2: We are grateful for the suggestion. To ensure greater clarity and address the reviewers' concerns, we have added detailed figure captions to the images, providing comprehensive explanations of the content. This change can be found-page3,section3,line69; page 8, section Ⅱ 2,line164; page 9, section Ⅱ 2,line171; page 10, section Ⅱ 2, line178; page 11, section Ⅱ 2,line185; page 18, section IV, line 433. 3. Comments 3: Numbering and references to figures: Check that the numbering of the figures matches the text. Sometimes, the figures are mentioned with different numbers or without a clear description;Ensure that there is an explicit reference in the text to each figure and table used Response 3:We sincerely apologize for the ambiguity in our previous description of the figures. In this revised version, we have numbered all the charts and referenced them at the pertinent points for improved clarity. This change can be found-page2, section 3, line 68; page3,section 4, line 84; page5,section 4, line 90; page14,section Ⅲ, line 300; page18,section Ⅳ, line 448. 4. Comments 4: Clarify whether the analysis is your own or based on existing studies. If the authors conducted the analysis, they must explain the methodology, indicating the database used, the criteria for data selection and the statistical approach. Indicate in the legend who the data originated from. The data must be clearly cited if they originate from other publications. Response 4:We apologize for the unclear data sources in the previous versions. We have provided annotations for all the data presented in the charts: the sources and analytical methods are specified in the figure captions, and references have been included following the relevant text. This change can be found-page6,section4, line94; page7,section Ⅱ 2,line159.
|
Reviewer 2 Report
Comments and Suggestions for Authors
The paper by Lu et al. explores p21-activated kinases (PAKs), their role in key physiological processes like cytoskeletal rearrangement and signal transduction, and their involvement in cancers, especially breast and pancreatic cancers. It emphasizes Group II PAKs (PAK4-6) in tumor progression and Group I PAKs (PAK1-3) in normal functions such as cardiovascular development and neurogenesis. The paper also covers the development of small-molecule inhibitors targeting PAKs and their challenges, particularly in achieving selectivity without affecting normal physiological roles.
Potential Questions:
- Could alternative modalities (e.g., PROTACs, RNAi, or CRISPR) overcome the limitations of traditional PAK inhibitors in selectivity and efficacy?
- Can we develop PAK inhibitors that target cancer-promoting functions without disrupting their physiological roles, or is this an inherent limitation of kinase inhibition?
- Visual Aids: Including a schematic overview or pictorial representation summarizing PAKs’ roles in both normal and tumor contexts would enhance clarity and impact.
Suggestions and Minor Comments:
- Capitalization and Formatting: Sentences, especially in page 2 table, don’t always start with capital letters. Consistent formatting will improve readability.
- Organization: The content could flow more logically. Clearer sectioning and structure would aid in understanding.
- Capitalization and Formatting: Sentences, especially in page 2 table, don’t always start with capital letters. Consistent formatting will improve readability.
- Organization: The content could flow more logically. Clearer sectioning and structure would aid in understanding.
Author Response
1. Summary |
|
|
We are deeply grateful for your insightful comments on our manuscript and extend our most respect for your expertise. As you rightly pointed out, there are several issues within our article that require addressing. In accordance with your suggestions, we have made revisions to the previous draft. The reviewers' comments are listed below in blue, with specific issues numbered. Our responses are provided in regular black font.
|
||
2. Questions for General Evaluation |
Reviewer’s Evaluation |
Response and Revisions |
Is the work a significant contribution to the field? |
|
|
Is the work well organized and comprehensively described? |
|
|
Is the work scientifically sound and not misleading? |
|
|
Are there appropriate and adequate references to related and previous work? |
|
|
Is the English used correct and readable? |
|
|
3. Point-by-point response to Comments and Suggestions for Authors |
|
|
1. Comments 1: Potential Questions: Could alternative modalities (e.g., PROTACS, RNAi, or CRISPR) overcome the limitations of traditional PAK inhibitors in selectivity and efficacy? |
||
Response 1: We appreciate the thoughtful and constructive review provided by the reviewers. Gene-level editing technologies have surmounted the constraints of conventional PAK inhibitors in terms of selectivity and efficacy. Nevertheless, they continue to encounter a range of challenges in practical applications, demanding continued rigorous research and technological refinement. Furthermore, in addition to playing a role in tumor development, PAK is also very important for the physiological functions of normal cells, so gene editing related technologies may not be very suitable for PAK. |
||
2. Comments 2: Can we develop PAK inhibitors that target cancer-promoting functions without disrupting their physiological roles, or is this an inherent limitation of kinase inhibition? |
||
Response 2: We thank the reviewers for the suggestions they offered us and have incorporated the recommended content into the discussion section of the manuscript. The challenges faced in developing PAK inhibitors that specifically target the functions driving cancer, without interfering with their physiological roles, arise not from the intrinsic limitations of kinase inhibition but from the inherent complexity of the PAK signaling pathway. While this presents a formidable obstacle, it remains a challenging yet attainable objective. This change can be found-page19,sectionⅣ,line 479. 3. Comments 3: Visual Aids: Including a schematic overview or pictorial representation summarizing PAKs’ roles in both normal and tumor contexts would enhance clarity and impact. Response 3: As the reviewer suggested, we have added a schematic diagram that outlines the roles of PAKs in both normal physiological and tumor environments. This change can be found-page18, section Ⅳ, Figure6. 4. Comments 4: Capitalization and Formatting: Sentences, especially in page 2 table, don't always start with capital letters. Consistent formatting will improve readability. Response 4: We are deeply sorry for the errors in the article format. We have standardized the case and format of the text in the table. This change can be found-page3, section 4, Table 1; page 4, section 4,Table 2; page 5, section 4,Table 3; page 14, section â…¢ 1, Table 4; page 14, section â…¢ 2,Table 5. 5. Comments 5: Organization: The content could flow more logically. Clearer sectioning and structure would aid in understanding. Response 5: We sincerely appreciate the reviewer for the constructive comment. In order to enhance the clarity of the article, we have refined and elevated the content and titles of the tables, image captions, and other elements, thus rendering the content and structure of each chapter more coherent and accessible.This change can be found-page3, section 4, line85; page 4, section 4,line87; page 5, section 4,line92; page 14, section â…¢ 1, line 303; page 14, section â…¢ 2,line307; page3,section3,line69; page 8, section â…¡2,line164; page 9, section â…¡ 2,line171; page 10, section â…¡ 2, line178; page 11, section â…¡ 2,line185; page 18, section IV, line 433.
|
Round 2
Reviewer 1 Report
Comments and Suggestions for Authors
The improvements in the presentation of tables with images have significantly increased clarity and readability. The correlation analysis is now much more precise. However, I suggest ensuring that all figures and tables are formatted consistently to maintain visual coherence throughout the manuscript. These revisions have strengthened the manuscript and made the presentation of the data more intuitive and the analysis more transparent.